# Targeting of the NRL Pathway as a Therapeutic Strategy to Treat Retinitis Pigmentosa

**DOI:** 10.3390/jcm9072224

**Published:** 2020-07-13

**Authors:** Spencer M. Moore, Dorota Skowronska-Krawczyk, Daniel L. Chao

**Affiliations:** 1Internal Medicine Residency, Department of Medicine, College of Medicine, University of Arizona, Tucson, AZ 85721, USA; spencer.m.moore@gmail.com; 2Department of Biophysics and Physiology, Center for Translational Research in Vision, Gavin Hebert Eye Institute, University of California, Irvine, CA 92602, USA; dorotask@hs.uci.edu; 3Andrew Viterbi Department of Ophthalmology, Shiley Eye Institute, University of California, La Jolla, CA 92602, USA

**Keywords:** retinitis pigmentosa, rod photoreceptors, gene therapy, mutation agnostic, optogenetics, regenerative medicine, neuroprotection, transcriptional regulation, NRL, NR2E3, rod cone conversion, homeostasis

## Abstract

Retinitis pigmentosa (RP) is an inherited retinal dystrophy (IRD) with a prevalence of 1:4000, characterized by initial rod photoreceptor loss and subsequent cone photoreceptor loss with accompanying nyctalopia, visual field deficits, and visual acuity loss. A diversity of causative mutations have been described with autosomal dominant, autosomal recessive, and X-linked inheritance and sporadic mutations. The diversity of mutations makes gene therapy challenging, highlighting the need for mutation-agnostic treatments. Neural leucine zipper (NRL) and NR2E3 are factors important for rod photoreceptor cell differentiation and homeostasis. Germline mutations in NRL or NR2E3 leads to a loss of rods and an increased number of cones with short wavelength opsin in both rodents and humans. Multiple groups have demonstrated that inhibition of NRL or NR2E3 activity in the mature retina could endow rods with certain properties of cones, which prevents cell death in multiple rodent RP models with diverse mutations. In this review, we summarize the literature on NRL and NR2E3, therapeutic strategies of NRL/NR2E3 modulation in preclinical RP models, as well as future directions of research. In summary, inhibition of the NRL/NR2E3 pathway represents an intriguing mutation agnostic and disease-modifying target for the treatment of RP.

## 1. Introduction

### 1.1. Introduction to Retinitis Pigmentosa

Retinitis pigmentosa (RP) encompasses a genetically heterogeneous collection of inherited progressive retinal degenerative disorders affecting approximately 1:4000 individuals [1] in the United States. Among the inherited retinal dystrophies (IRDs), RP is the most common. Shortly after German physician and scientist Hermann von Helmholtz invented the direct ophthalmoscope in the 1850s, the first descriptions of RP were recorded [2]. Despite its genetic heterogeneity, RP is marked clinically by retinal findings of bone-spicule pigmentary clumping, vascular attenuation, and optic nerve (ON) pallor [3]. RP may present as part of a systemic syndrome (termed syndromic RP) [4], accounting for 20% of cases or with only ocular manifestations (non-syndromic RP, accounting for up to 80% of cases) [5]. Patients usually experience symptoms of nyctalopia and progressive concentric peripheral visual field loss beginning as early as adolescence or young adulthood, with progressive visual decline. In RP, there is a progressive loss of rod photoreceptors, which is followed by a secondary cone photoreceptor degeneration, at which point patients experience decreased central visual acuity. Developing effective treatments for IRDs including RP remains challenging, particularly due to the genetic heterogeneity underlying the disease.

### 1.2. Genetic Heterogeneity of Retinitis Pigmentosa

To date, nearly 100 unique genes have been implicated in RP pathogenesis, with significant overlap among genes shared with RP and the related IRDs including Leber’s Congenital Amaurosis (LCA), cone-rod dystrophy, and macular dystrophy [3]. Inheritance patterns vary from autosomal dominant [6,7] (15–35%) and autosomal recessive [8,9] (15–20%), to X-linked [10,11] (10–15%) and sporadic (30%) [11]. Mutations in the NRL and NR2E3 genes encoding rod-specific transcription factors exemplify this diversity, as autosomal dominant [12,13], autosomal recessive [14,15], and sporadic [16] mutations have been implicated in RP and the related IRD, enhanced S-cone syndrome (ESCS). In addition to heterogeneity in inheritance patterns, prevalence of individual mutations is quite low, with no single mutation or locus accounting for >10% of identified cases, and most mutations accounting for 1% or fewer of cases [3,5]. Despite the considerable genetic heterogeneity in RP pathogenesis, investigation of specific RP-associated genes has yielded important insights into critical visual cycle-associated proteins as well as structural and metabolic proteins involved in rod photoreceptor development and function, including rhodopsin [17], RPE65 (expressed in retinal pigmented epithelium and supports phototransduction) [18], RPGR [19] and NRL/NR2E3 [20,21]. Some of the most commonly detected gene mutated in RP include USH2A, RPGR, EYS, RHO, RP1, and RPGR [3], and autosomal recessive cases are generally diagnosed earlier in life, as predicted by Mendelian inheritance patterns [22]. Gene augmentation strategies of select mutated genes as a therapy for inherited retinal diseases have been the subject of extensive investigation. Clinical trials delivering an adeno-associated viral vector (AAV) containing the wild-type human RPE65 cDNA in patients with RPE65-associated Leber’s Congenital Amaurosis have demonstrated long term improvement in visual function [23,24]. In 2017, the FDA formally approved voretigene neparvovec-rzyl (Luxturna, Spark Therapeutics), as the first gene therapy for an IRD. Similar gene therapy trials using a gene augmentation strategy for other IRDs with single mutations are underway or currently are recruiting [25]. Despite the exciting progress in gene therapy for RP and IRDs, this approach may not be amenable for many mutations associated with RP. First, packaging of replacement genes into AAV must be individualized for each disease, which is impractical from a regulatory and clinical development perspective for the large number of genetic mutations that cause RP [3]. Second, viral vectors are limited in carrying capacity (maximum of 4.8 kb for recombinant AAV) of cDNA, thus excluding larger genes or requiring a complex multi-vector strategy [26]. Third, gene augmentation strategies using gene therapy are best suited for recessive alleles and less ideal for most autosomal dominant mutations that result in a dominant negative phenotype (except for those amenable to expression of the wt protein). Thus, developing a therapy that would work for all types of RP, independent of the type of mutation is of great interest and is an unmet need in the field.

### 1.3. Toward a Mutation-Independent Treatment

Conceptually, therapeutic strategies that would be effective across all forms of RP can be divided into three broad categories: (1) neuroprotective strategies to prevent photoreceptor death; (2) regenerative medicine approaches to replace lost photoreceptors or prevent photoreceptor death; and (3) optogenetic approaches to endow cells that are normally not light sensitive (e.g., retinal ganglion cells) and are not affected by RP to be sensitive to light by gene therapy or chemical means. Unfortunately, there have been no successful Phase 3 trials using these strategies thus far (Table 1). Ciliary neurotrophic factor (CNTF) was the first neuroprotective agent to progress to clinical trials, based on extensive preclinical evidence for slowing retinal degeneration in animal models [27]. Although an intraocular CNTF-releasing implant was shown to be safe in a Phase 1 trial [27], the treatment was shown to be inferior to the sham control eye in a Phase 2 trial for RP [28]; adaptive optics scanning of CNTF implant-treated patients did show significantly thicker outer retina layers and slowed decline in cone density despite no changes in visual acuity, visual field sensitivity, or ERG [29]. Another neurotrophic factor, nerve growth factor (NGF), showed no significant adverse effects in a topical preparation administered to RP patients, and a minority of patients reported subjective visual benefits in a pilot trial [30]. A Phase 2 trial investigating oral valproic acid as a neuroprotective agent for RP recently reported negative results versus placebo [31]. The antioxidant molecule n-acetylcysteine (NAC) recently showed promise in cone photoreceptor function in a Phase I trial for RP [32]. Another intriguing neuroprotective factor, rod-derived cone viability factor (RdCVF), has shown promise in protecting cones from oxidative stress-mediated degeneration in preclinical RP models [33]. Other small molecules or neuroprotective treatments reaching clinical trials in RP included brimonidine, lutein, and vitamin A and vitamin E, though none have demonstrated significant efficacy in clinical trials [34]. 

Surgical cell/tissue transplantation generally aims to replace degenerated photoreceptors or RPE from various donor sources. This approach is thought to either replace dying photoreceptors or RPE or secrete neurotrophic factors which may prevent photoreceptor death. Preclinical models of stem cell-derived retinal cells have shown early promise that has not yet been demonstrated in human clinical trials. When human embryonic stem cells (hESC) or induced pluripotent stem cell (iPSC)-derived photoreceptors were transplanted in retinal degeneration *Pde6β*^-/-^ mice there was preliminary evidence of functional photoreceptor integration into the mouse retina [35]. Intravitreal injection of autologous bone marrow-derived stem cells showed no long-term benefit in RP [36]. Previous trials of fetal RPE grafts for age-related macular degeneration (AMD) also have not shown efficacy and some patients suffered issues with immune rejection [37]. Fetal tissue grafts have not been extensively studied in RP other than a single patient reported in 2004 [38]. Several clinical trials currently recruiting or underway will investigate allogeneic grafts of human embryonic stem cell (hESC)-derived retinal or neural progenitor cells in RP (NCT02384293, NCT02464436, NCT03073733, NCT03944239). One such trial sponsored by jCyte showed promising safety results for a single intravitreal injection of its jCell human retinal progenitor cells in a Phase 1/2a trial (NCT02320812); the Phase 2b is currently underway (NCT03073733).

A strategy termed optogenetics has emerged as a promising strategy for conferring light sensitivity into the surviving non-photoreceptor cells in retinal degenerative disease. This can occur either through the genetic introduction of light-sensitive ion channels such as channelrhodopsin, or through small molecule approaches using light-sensitive small molecules which can open ion channels to depolarize neurons [39,40]. Channelrhodopsins are light-gated ion channels originally discovered in unicellular algae that can confer the ability of neurons to depolarize in response to certain wavelengths of light [41]. Several clinical trials currently recruiting or underway package channelrhodopsin DNA into viral vectors for intravitreal injection for delivery to the retina (NCT04278131, NCT03326336, NCT02556736). Another approach to render cells light-sensitive is the use of photoswitches, small molecules that can change conformation at certain wavelengths of light, in turn opening or closing ion channels. Preclinical rodent studies have demonstrated that these can confer light sensitivity to retinal ganglion cells and restore light sensitivity in animal models of RP [40]. Although an intriguing means of bypassing diseased retinal tissues, the visual resolution outcomes of such an approach are unproven and are unlikely to approach native retina. Retinal prostheses are conceptually similar in that they confer light sensitivity by transmitting visual stimuli to intact tissues, thus bypassing the degenerating photoreceptors [42,43]. Perhaps the best-known prosthesis, the Argus II, is a surgical intraocular implant that transduces visual stimuli to the retina from an external camera. 

Mutation-independent treatments are attractive for the prospect of treating greater numbers of RP patients with a single generic treatment. At present, however, none specifically target the pathologic biology of photoreceptor degeneration. A greater understanding of photoreceptor molecular biology is likely to yield additional clinically relevant insights. To this end, the transcription factor NRL and orphan nuclear receptor NR2E3 transcription factors are the subject of extensive investigation in photoreceptor transcriptional regulation [20,44,45] and represent a novel mutation-agnostic therapeutic strategy for RP treatment. 

## 2. Main Text

The NRL/NR2E3 pathway has been implicated in differentiation and subsequent maintenance of rod photoreceptors throughout the lifetime of the cell. Mice which lack NRL or NR2E3 activity do not have rods and instead have an increased number of cone-like cells expressing S-opsin. Physiologically, human patients with mutations in NRL/NR2E3 have phenotypic similarities in reduced or absent rod function with increased S-cone function. This suggests that blocking the NRL/NR2E3 pathway may endow rods with certain cone-like features that may prevent cell death by reducing or abolishing the manifestation of rod degenerative phenotypes. Here we review the scientific premise and evidence behind this therapeutic strategy. 

### 2.1. Human Phenotypes of NRL Pathway Mutations

Human mutations in the NRL/NR2E3 pathway leading to RP phenotypes provide proof of principle for the role of this pathway in photoreceptor development and homeostasis. Mutations in NRL have been implicated in both autosomal recessive (A76V) [15] and autosomal dominant RP [12,13,46,47]. Interestingly, all NRL mutations associated with autosomal dominant RP are missense mutations in only three residues, amino acids 49-51 [48]. Patients’ central acuity ranged from 20/20 to 20/200 with severely reduced ERGs and constricted visual fields. On the fundus exam, patients were noted to have pigment clumping and attenuated vessels with vision loss beginning in the first decade of life. These mutations are thought to be a gain of function mutations, as these mutations have been shown to lead to increased Rho expression in in vitro assays [49,50].

There are few reports of autosomal recessive RP due to NRL, with one report describing patients with homozygous mutations in both NRL as well as PABPN1, a cause of oculopharyngeal muscular dystrophy [15,51,52]. The most complete clinical description was in a pair of siblings found to have two allelic mutations of NRL: a nonsense mutation and a missense mutation in a critical binding site thought to be pathogenic. In vitro data suggest that this was a severe loss of function mutation. Only one of the patients was examined clinically at age 51. He was found to have relatively good central visual acuity (20/40 in his non-amblyopic eye) and reported night vision loss in early childhood. On fundus exam, the patient had abnormal pigment clumping in the periphery termed clumped pigmentary retinal degeneration [53] and attenuated vessels and had constricted visual fields on static perimetry. The patient also had a severe reduction in rod and cone function by ERG; however, the patient had normal color vision testing, with a slight enhancement of S-cone activity by visual fields [15]. Enhancement of S-cone activity was also found on ERG in the other clinical description of patients homozygous for NRL and PABPN1 [51], mirroring the *Nrl^-/-^* mouse [20]. 

Recessive and dominant mutations in human NR2E3 have also been associated with a wider spectrum of retinal disease. Pathogenic NR2E3 mutations have been associated with enhanced S-cone syndrome (ESCS)/Goldman-Favre syndrome [14,54], clumped pigmentary retinal degeneration, as well as autosomal recessive [55] and autosomal dominant RP [56]. In support of the theory of NR2E3 in suppressing cone development, autosomal recessive mutations in human NR2E3 have been linked to ESCS. Enhanced S-cone syndrome is a slowly progressive retinal degeneration with patients usually reporting nyctalopia in the first decade of life. Patients have a variable fundus image, as well as variable visual acuity, usually secondary to the presence of cystoid macular edema or macular schisis. This disease is diagnosed through pathognomonic features on ERG: loss of rod response and increased response to short-wavelength cones [14,54]. Of the three predominant human cone types, S-cones are the most sensitive to short-wavelength light due to production of S-opsin pigment. ESCS patients harboring NR2E3 mutations demonstrate reduced rod and L- and M-cone sensitivity but enhanced S-cone sensitivity on visual field testing and exaggerated ERG response to short-wavelength light [14,54]. On postmortem analysis of the degenerated retina of an ESCS patient harboring a homozygous autosomal recessive NR2E3 mutation (R311Q), investigators found absent rods and increased cones primarily of the S-cone type [44]. NR2E3 mutations were proposed to cause loss of normal retinal laminar architecture due to disproportionate levels of S-cones [45]. Longitudinal data suggests that patients with mutations in NR2E3 may have preserved central acuity, or milder progression of disease compared to other retinal degenerations unless macular schisis or cystoid macular edema is present [54,57]. In one unusual case, no known NR2E3 mutations were detected in an atypical ESCS patient, although a heterozygous-predicted loss-of-function NRL mutation was detected, suggesting a potential phenotypic NR2E3–NRL overlap [58]. Taken together, NRL/NR2E3 pathway mutations alter the balance of normal photoreceptor differentiation in the developing retina, leading to a loss of rod function and retinal degeneration. Importantly, deleterious phenotypes such as retinal degeneration seen in human patients with germline mutations of NRL or NR2E3 have not been observed in long-term deletion of NRL in mature retina in mouse models. A pragmatic therapeutic goal for RP patients would be the reprogramming of rod photoreceptors to avoid manifestations of retinal degeneration, rather than phenocopying germline NRL or NR2E3 mutations. Thus, mutations in NRL and NR2E3 highlight actionable underlying photoreceptor biology, and extensive basic science evidence supports the concept of manipulating rod transcriptional regulation as a neuroprotective strategy. 

### 2.2. NRL as a Regulator of Rod Photoreceptor Transcription

Mammalian photoreceptors are a type of specialized postmitotic neuroepithelial cell originating from a common photoreceptor precursor, generating the first cones by 8 weeks’ gestation and first rods by 10 weeks’ gestation in the developing human retina [59]. Recent work has shed light on the developmental transcriptional program regulating the commitment of photoreceptor cell fate (Figure 1). NRL encodes a Maf-family basic motif-leucine zipper transcription factor NRL that is exclusively expressed in developing, as well as mature, rod photoreceptors in the retina [60,61]. Germline NRL deletion resulted in 161 genes differentially expressed between wt and NRL^-/-^ retina, with gene ontology analysis revealing that affected genes were involved in signal transduction and transcriptional regulation, including Rho [62,63]. In 2001, Swaroop’s group reported that NRL was required for rod photoreceptor development in the mouse retina [20]. Interestingly, electroretinogram (ERG) recordings from their NRL^-/-^ mouse revealed absent rod activity and abnormally high S-cone activity (i.e., the default photoreceptor fate); the rod pigment rhodopsin could not be detected immunohistochemically, leading the authors to conclude that NRL was critical for differentiation of rod photoreceptors. NRL absence favored development of rods with cone-predominant physiology, including cone-like nuclear morphology and increased blue light-adapted ERG cone activity, suggesting unexpectedly elevated cone-like light response in the retina [20]. Indeed, germline NRL deletion, which confers rods with S-cone-like properties, somewhat resembles NR2E3 recessive mutations in humans causing ESCS, again highlighting the phenotypic overlap among these transcriptional regulators [14,20]. Evidence demonstrated that NRL acts as an initial transcriptional switch promoting rod development from undifferentiated photoreceptor precursors [64]; importantly, this master transcriptional switch was shown to alter hundreds of non-coding RNAs [65] and appears to act in mice between P6-P10, corresponding with photoreceptor precursor commitment to the rod fate [66]. Thus, NRL was established as a master transcriptional regulator essential for rod photoreceptor development. 

### 2.3. NR2E3 Suppresses Cone Transcription

NR2E3 encodes the orphan nuclear receptor NR2E3 specific to rod photoreceptor cell nuclei that is conserved among vertebrates, where it suppresses transcription of cone-specific genes in developing rod photoreceptors [21] (Figure 1). The rd7 retinal degeneration mouse originally described in 2000 later had its mutation mapped to NR2E3 [67]. In this mouse, rod precursors express cone genes, suggesting the S-cone fate as the default in the absence of NRL/NR2E3 signaling [68]. NRL directly binds regulatory regions of NR2E3 [69], and NR2E3 expression is absent in NRL^-/-^ mice [20], suggesting that NR2E3 is a downstream transcriptional target of NRL. Chen and colleagues proposed a signaling model in which NRL acts in rod photoreceptors to activate transcription of rod genes, including NR2E3, and to repress cone gene transcription [21]. Ectopic expression of NR2E3 was sufficient to bias retinal progenitors toward rod-like fate, in both the NRL^-/-^ mouse retina, as well as in wildtype Xenopus retina [70,71]. NR2E3 expression was localized to the outer nuclear (photoreceptor cell body) layer (ONL) of the retina and most abundantly following rod development, suggesting it functions to suppress cone gene expression [72]. Indeed, expression levels of various rod- and cone-specific genes were shown to be altered in the retina of NR2E3-deficient mice [73]. Under the control of the NRL promoter, NR2E3 expression in rd7 NR2E3^-/-^ mice prevented retinal degeneration, supporting the hypothesis of NR2E3 favoring rod homeostasis [68]. Thus, evidence suggests that in the developing retina, NR2E3 represses cone development among mitotic photoreceptor progenitors, while maintaining the rod transcriptional profile in mature rods [74] as a downstream target of NRL.

### 2.4. Manipulating the NRL Pathway as a Neuroprotective Strategy in RP

Germline mutations in NRL or NR2E3 cause RP by perturbing the normal balance of photoreceptor cell fate and interfering with rod homeostasis in the mature retina. One potential therapeutic strategy is that inhibition of NRL or NR2E3 expression may reduce rod gene expression and/or disinhibit cone transcription, thus reducing the potential for disease manifestation (Figure 2).

Multiple efforts from independent investigators have validated this proof-of-concept approach in preclinical rodent RP models (Table 2). Montana et al. published their report targeting NRL in the mature retina. In a mouse with tamoxifen-inducible Cre recombinase expression and floxed NRL alleles, NRL knockout in the mature mouse retina led to endowing rods with certain features specific to cones [75]. These included: (1) upregulation of certain genes expressed in cones and downregulation of genes expressed in rods; (2) morphologic features including enlarged nuclei found on electron microscopy consistent with cone morphology; and (3) electrophysiologic characteristics of cones including enhanced photopic a-wave responses, desensitization of scotopic responses and rapid inactivation of photoresponses, and recovery of the photoresponse in the presence of 9-cis-retinol [75]. Importantly, there did not appear to be any change in the wiring of the rods, increased ERG response to short wavelengths, or retinal degeneration, which occurs in germline NRL^-/-^ mice. When the adult NRL knockout experiment was repeated on the Rho^-/-^ retinal degeneration mouse model at P25, the knockout prevented retinal degeneration, preserved retinal laminar architecture, and restored photopic cone ERG physiology when assayed at P90 [75]. This suggested that deletion of NRL in the adult mouse could prevent degeneration in a rodent model of RP. More recently, the CRISPR/Cas9 gene-editing system has gained attention for gene knockout in adult tissues. Yu and colleagues reported an AAV packaging and delivery system for CRISPR/Cas9 which could effectively transduce mouse photoreceptors via subretinal injection [76]. A small guide RNA (sgRNA) construct against NRL successfully reduced out NRL protein expression from photoreceptors. On gene expression analysis, only 147 of the 6000 differentially regulated genes between rods and cones significantly changed. NRL knockout caused downregulation of rod-associated genes including NR2E3 needed for rod phototransduction. In the absence of NRL, rods assumed the cytologic and chromatin architecture features reminiscent of cones. On ERG, scotopic rod function was depressed but cone function was unaffected [76]. Importantly, adult CRISPR/Cas9 NRL knockout mice lacked the deleterious phenotypes of retinal lamination, retinal degeneration, and vascular and RPE changes characteristic of NRL germline knockout. In Rho^-/-^ germline knockout mice, CRISPR/Cas9 NRL knockout at P14 or P28 significantly slowed the loss of ERG light-adapted response and increased the ONL surviving photoreceptor density. In the Rd10 rod degeneration mouse model caused by mutation in Pde6β, CRISPR-Cas9 NRL knockout at P14 prevented both rod and cone degeneration and improved light-adapted ERG response. Finally, CRISPR/Cas9 NRL knockout at P14 in the Rho^P347S^ rod degeneration model significantly slowed light-adapted b-wave ERG degeneration over time and protected photoreceptors from apoptosis [76]. A double sgRNA strategy targeting two distinct NRL or NE2E3 sites simultaneously via CRISPR/Cas9 knockout at P7 revealed increased expression of the cone marker mouse cone arrestin when injected in wild-type (wt) mice and increased ONL thickness in 2 different Pde6β^-/-^ retinal degeneration mouse models [77]. ERG photopic response was also increased, indicative of increased cone function [77]. Thus, in multiple mouse models, adult knockout of NRL appears to slow the rate of retinal degeneration and preserve crucial aspects of visual-cycle physiology, suggesting a potential target for further exploration in human RP. Importantly, deletion of NRL in adult photoreceptors did not appear to cause deleterious effects such as photoreceptor degeneration even after six months, suggesting that this therapy may not cause retinal degeneration [75,76]. This suggests that NRL deletion in adult mice is fundamentally different to deleting NRL in development, in that there is not a full conversion of rods to S-cones. Instead, rods gain a subset of properties found in cone photoreceptors, which somehow leads to prevention of rod photoreceptor death in RP models. This may also explain why there is not photoreceptor degeneration when NRL is deleted in adult compared to germline NRL^-/-^ mice. Moreover, epigenetic modifications to photoreceptor genes made in development may render adult cells less flexible to cone reprogramming versus the developing retina.

### 2.5. Manipulating the NR2E3 Pathway as a Neuroprotective Strategy in RP

Unlike NRL, NR2E3 inhibition strategies include both small-molecule and genetic modalities (Table 2). Nakamura and colleagues screened for small molecule inhibitors of NR2E3 [78], and their candidate molecule photoregulin-1 (PR1) reduced expression of rod genes Rho, Nrl, Gnat1, and NR2E3 and rhodopsin protein in developing mouse retinal explants; cone gene (Thrb) and protein (S opsin) expression was increased. In vivo, PR1 intravitreal injection prevented photoreceptor death in both the Rho^P23H^ rod degeneration mice and in the Pde6b^rd1^ rod degeneration mouse [78]. Thus, the NR2E3 antagonist PR1 provided preliminary evidence of photoreceptor neuroprotection, although the authors did not demonstrate effects on visual physiology in vivo. The same group later published their findings on a next-generation NR2E3 inhibitor, photoregulin-3 (PR3), which reduced rod-specific gene expression and increased S-opsin^+^ cells in culture and in vivo following systemic PR3 treatment of wt mice [79]. Systemic treatment of PR3 prevented rod photoreceptor death in the Rho^P23H^ rod degeneration mouse and also improved photopic and scotopic function on ERG [79]. Although the authors assayed retina histology, gene expression, and ERG after only 1 week of treatment, the potential of NR2E3 inhibition to slow retinal degeneration in vivo represents an intriguing possibility for treatment of RP. 

The benefits of genetic manipulation of the NE2E3 are less clear. Naessens et al. provided proof-of-concept of an antisense oligonucleotide capable of NE2E3 knockdown in vitro but did not investigate functional consequences [80]. Interestingly, a report using a gene therapy NR2E3 overexpression strategy also showed promise in mouse retinal degeneration models [81]. Wild-type NR2E3 cDNA was packaged into an AAV8 vector and injected subretinally in various mouse retinal degeneration models; AAV8-Nr2e3 treatment at P0 preserved ONL photoreceptor density, increased green and blue opsin^+^ and rhodopsin^+^ cells, and improved ERG response, which the authors attributed to increased recruitment of phototransduction-relevant transcription factors in a number of rodent preclinical models of RP [81]. This suggests that modulation of the NR2E3 pathway (either inhibition or upregulation) in mouse models of RP may therefore slow retinal degeneration by either similar or distinct mechanisms. NR2E3 antagonism may reduce rod photoreceptor susceptibility to genetic retinal degeneration by conferring cone-like properties [79], whereas NR2E3 overexpression may reset rod gene expression networks and improve retinal homeostasis [81].

## 3. Conclusions/Future Directions

RP presents a significant multifaceted challenge for developing therapeutics in ophthalmic genetics. Genetic heterogeneity among diagnosed cases complicates gene therapies and favors the most prevalent autosomal recessive mutations that are amenable to gene replacement drug development. However, patients harboring less prevalent mutations or mutations with dominant inheritance are unlikely to benefit from currently available gene therapies. Thus, recent research has focused on mutation-agnostic treatments. Channelrhodopsin gene therapies under investigation look to confer photoreceptor properties on retinal cells such as interneurons or retinal ganglion cells that do not degenerate in RP although with likely poor visual sensitivity and resolution. Clinical trials of neuroprotective small molecules have yielded disappointing results, and to date none have gained FDA approval. 

In the natural history of RP progression, initial rod dysfunction and degeneration precedes later cone degeneration and worsening loss of central vision [3]. Preventing cone photoreceptor degeneration is therefore of great therapeutic interest. Work on the NRL/NR2E3 nuclear receptor pathway has identified an intriguing mutation-agnostic and potentially disease-modifying therapeutic target for RP. Preclinical work from multiple independent groups has validated inhibition of the NRL pathway as a promising mutation agnostic therapy for RP. These include several different but complementary methods of NRL pathway inhibition including inducible gene knockout, CRISPR/Cas9 gene editing, as well as small molecule inhibition, strengthening the scientific validity of NRL as a therapeutic target for RP.

What might be the molecular mechanisms behind photoreceptor survival when inhibiting the NRL pathway? This remains an unanswered question in the field. One possibility is that decreased expression of rod genes such as rhodopsin, whose protein levels are exquisitely regulated to prevent photoreceptor degeneration [82], may decrease metabolic stress and allow a steady state “retinal homeostasis”, which may prevent photoreceptor degeneration. Another possibility could be the upregulation of neuroprotective factors preventing photoreceptor death. However, significant upregulation of established protective cytokine or neurotrophic pathways such as STAT3 or CNTF, respectively, have not been observed in NRL^-/-^ rods compared to controls [76], but low level expression of multiple factors acting in synergy cannot be excluded. 

One other unanswered question is whether long term inhibition of the NRL pathway will lead to deleterious effects on the retina such as retinal degeneration. NRL^-/-^ germline knockout mice have abnormal retinal lamination, photoreceptor death, as well as Müller glia dysfunction and changes in vascular permeability [83]. However, none of these changes were observed when the NRL pathway was inhibited in adult mice for up to seven months after NRL knockout [76], suggesting that inhibition of the NRL pathway in adults may not lead to the deleterious phenotypes present in germline knockouts of NRL in mouse models of RP in human patients with NRL mutations. These phenotypes could be due to the role of NRL during retinal development, rather than the effects of NRL during adulthood. Indeed, adult photoreceptors possess limited plasticity, which is supported by the fact that only a small subset (about 2% comparing adult to germline NRL knockout) of differentially regulated genes between rods and cones are significantly changed in NRL^-/-^ photoreceptors [76]. Therefore, it is likely that the deletion of NRL in adult mice results in only limited changes to rod gene expression, which provides neuroprotective properties but does not disrupt metabolic homeostasis or axon wiring of the photoreceptors. Of course, the native human retina best serves visual sensitivity and resolution with a combination of both rod and cone photoreceptors, and therapeutic reprogramming of rods in the context of RP would diminish any remaining rod function while also modifying cone function, with uncertain effects on visual perception in patients. However, the risks of altering the nature of visual perception would likely be less than the risks of progressive retinal degeneration in RP. Additional questions include: what is the optimal timing of NRL deletion during retinal degeneration? Moreover, does deletion of NRL prevent photoreceptor degeneration in large animal models of RP? In addition, are there certain imaging biomarkers (e.g., optical coherence tomography) or genomic biomarkers that may predict the response to this therapeutic approach? We eagerly await further experiments to answer these questions. 

In conclusion, inhibition of the NRL/NR2E3 pathway represents an intriguing approach for the treatment of retinitis pigmentosa. It represents an attractive target of future investigation for its potential to modify the natural history of RP retinal degeneration from multiple genetic etiologies. We hope that further research will refine this approach and allow for testing of this hypothesis in clinical trials, which brings the possibility of preserving sight for the many patients with untreated retinal degenerative diseases.

## Figures and Tables

**Figure 1 jcm-09-02224-f001:**
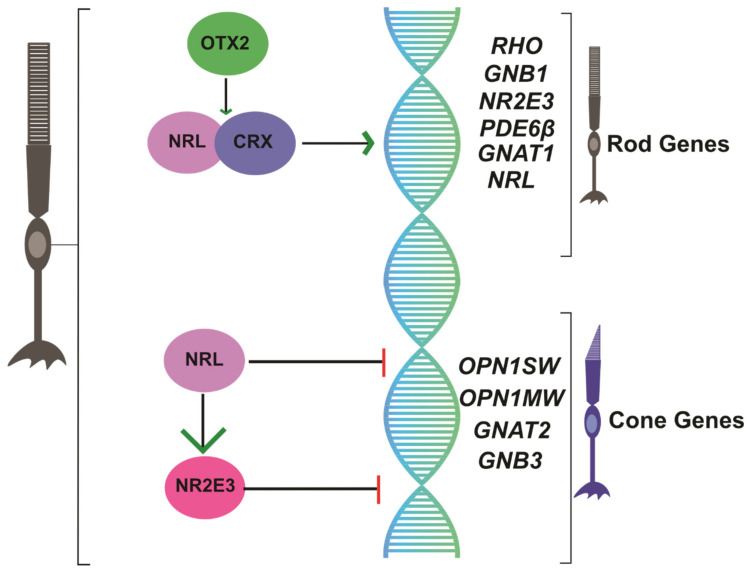
Schematic of NRL/NR2E3 regulation of photoreceptor gene transcriptional regulation in rod cells.

**Figure 2 jcm-09-02224-f002:**
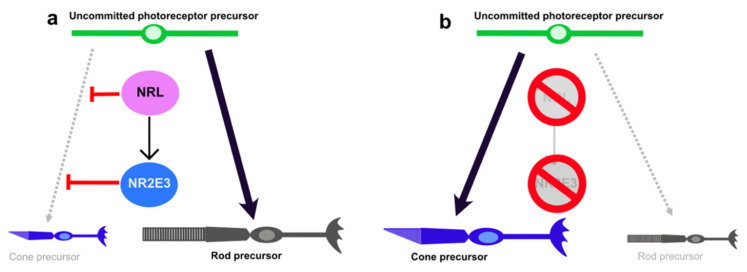
NRL pathway in rod photoreceptor differentiation. (**a**) In normal situations, NRL acts through NR2E3 to inhibit cone photoreceptor fate and promote rod photoreceptor fate. (**b**) Deletion of NRL or NR2E3 promotes a cone precursor fate.

**Table 1 jcm-09-02224-t001:** Pros and cons of mutation-agnostic treatment strategies in RP.

Treatment Strategy	Advantages	Disadvantages	Clinical Trial Examples	Noteworthy Outcomes
Neuroprotection	-Simplicity of drug delivery/noninvasive -FDA-approved agents	-Unproven in neurodegenerative disease -Importance of timing early in disease process	-CNTF [27,28,29] -NGF [30] -Valproic acid [31] -NAC [32] -RdCVF [33]	-CNTF intraocular implant: inferior to sham control -Topical NGF: vision improved in minority of patients -PO valproic acid: negative results vs. placebo
Regenerative medicine/cell transplant	-Theoretical capacity to replace lost cells/tissues -Future ability to reprogram patient-derived stem cells to avoid immune rejection	-May require surgical intervention -Uncertain potential for cellular integration and function -Potential for immune rejection in allografts	-Bone-marrow derived stem cells [36] -Human embryonic stem cell derived retinal/neural progenitor cells	-No long-term benefits reported from BMDSCs -hESC retinal/neural progenitor cell transplants recruiting/underway (NCT02384293, 02464436, 03073733, 02320812)
Optogenetics/prostheses	-Ability to increase light sensitivity	-Poor visual resolution -Unproven durability	-Channelrhodopsins [39,40,41] -Photoswitches [40] -Retinal prostheses [42,43]	-Channelrhodopsin trials recruiting/underway -Argus II retinal prosthesis– improved 5yr visual function -Alpha IMS retinal prosthesis recent CE approval in EU
Photoreceptor reprogramming	-Conferring resistance to degenerative photoreceptor loss	-Modification of intrinsic visual physiology -Potential changes in visual perception -Requires timely intervention		

**Table 2 jcm-09-02224-t002:** Effects of NRL/NR2E3 pathway manipulation on vision and retinal degeneration.

Study	Model(s)	Experimental Manipulation	Timing	Effects on Visual Physiology	Effects on Retinal Degeneration	Effects on Gene/Protein Expression
Mears 2001 PMID 11694879 [20]	wt mice	Germline *Nrl* deletion	Germline knockout	-Absent ERG scotopic rod response -2-3x increased ERG photopic cone response -6x increased ERG 400nm S-cone response	-ONL photoreceptor nuclei appear cone-like, formation of rosette-like structures	-Absent *Nr2e3*, *Pdeb*, *Rho*, *Gnat1* (rod gene) expression -Increased *Opn1sw*, *Gnat2*, *Car* (cone gene) expression
Montana 2013 PMID 23319618 [75]	-*Nrl^fl/fl^* CAG-Cre -*Nrl^fl/fl^* CAG-Cre, *Rho^-/-^* germline mice	-Tamoxifen injection inducing Cre-recombinase expression	-KO P42, analysis P63 (*Nrl^fl/fl^* CAG-Cre) -KO P25-P28, analysis P90 (*Nrl^fl/fl^* CAG-Cre *Rho^-/-^*)	-*Nrl* KO: significantly decreased scotopic (rod) and increased photopic (cone) function in vivo; 35x desensitization and rapid inactivation of photoresponse -*Rho^-/-^ Nrl* KO: increased photopic ERG cone function	-*Nrl* KO: variable ONL waviness, no rosettes -*Rho^-/-^ Nrl* KO: increased ONL cell density, cone opsin expression	-*Nrl* KO: absent *Nr2e3*, *Rho*, *Gnat1*, *Gnb1* (rod gene) expression; increased *Gnat2*, *Gnb3* (cone gene) expression
Yu 2017 PMID 28291770 [76]	-wt mice -*Rho^-/-^* -*Pde6β^-/-^* -*Rho^P347S^*	-Subretinal injection of AAV8-CRISPR/Cas9 *Nrl* sgRNA construct	-Injection P14, analysis P90-P105	-wt *Nrl* KO: decreased scotopic ERG rod function, stable ERG cone function -*Rho^-/-^ Nrl* KO: slowed decline in photopic ERG function -*Pde6β^--/-^ Nrl* KO: preserved photopic ERG function -*Rho^P347S^ Nrl* KO: slower decline of photopic ERG function, improved optomotor response	-wt *Nrl* KO: no significant retinal structure changes -*Rho^-/-^ Nrl* KO, *Pde6β^--/-^ Nrl* KO, *Rho^P347S^ Nrl* KO: preserved ONL cell density	-wt *Nrl* KO: mild decreased rod gene expression, increased *Gnb3*, *Arr3* (cone) gene expression -*Rho^-/-^ Nrl* KO: increased S-opsin staining -*Pde6β^--/-^ Nrl* KO: increased cone arrestin, S-opsin staining - *Rho^P347S^ Nrl* KO: increased S-opsin staining
Zhu 2017 PMID 28429769 [77]	-wt mice- *Pde6β^-/-^*	-Subretinal injection of AAV-CRISPR/Cas9 *Nrl* or *Nr2e3* double sgRNA construct	-Injection P7, immunohistochemistry P30 or P50, ERG P50 or P60	-*Pde6β^-/-^ Nrl* or *Nr2e3* KO: increased photopic ERG function, no effect on scotopic ERG a-wave, small increase in scotopic ERG b-wave	-*Pde6β^-/-^ Nrl* or *Nr2e3* KO: increased ONL thickness	-wt and *Pde6β^-/-^* *Nrl* or *Nr2e3* KO: increased cone arrestin^+^ cells in ONL
Haider 2000 PMID 10655056 [68]	Human ESCS patients with *NR2E3* mutations	none	Germline	-12 degree visual field testing: decreased sensitivity to rod and L/M cone stimuli, 30x increased sensitivity to S-cone stimuli -increased ERG response to 450nm stimulus	-variable OCT abnormalities including foveal cysts	
Haider 2001 PMID 11487564 [14]	wt mice	*Nr2e3* knockout	Germline		-Disrupted ONL, whorl formation -Increased cone cells, including 203x increased blue-opsin^+^ cones	See Corbo and Cepko 2005 PMID 16110338 – hybrid rod-cone gene expression in *Nr2e3^-/-^* mouse retina
Milam 2002 PMID 11773633 [44]	Human ESCS patients with *NR2E3* mutations	none	Germline	-Visual field testing: supranormal S cone function and S:L/M cone function	-Fewer layers of ONL photoreceptor nuclei -No rods detected -Increased S-opsin staining	
Nakamura 2017 PMID 29148976 [79]	-wt mice -*Rho^P23H^*	Intraperitoneal injection of Nr2e3 inhibitor PR3	Injection P12-P14 or P21, analysis P14 or P21	-*Rho^P23^*^H^: significantly increased scotopic and photopic ERG function	-wt: increased S-opsin^+^ cells, truncated photoreceptor outer segments -*Rho^P23H^*: protection against ONL photoreceptor loss/increased ONL thickness	-wt: reduction of rod gene expression, unchanged cone gene expression -*Rho^P23H^*: increased Rcvrn and Rho expression
Li 2020 PMID 32123325 [81]	-wt mice -*Pde6β^--/-^* -*Rho^-/-^* -*Rho^P23H^* -*Cep290^-/-^* -*Nr2e3^-/-^*	Subretinal injection of AAV8-*Nr2e3* overexpression vector	-Injection P0, analysis P30 or P90-P120 -Injection P21, analysis P80-P110	-wt mice: no ERG changes -*Pde6β^-/-^, Rho^-/-^, Rho^P23H^, Cep290^-/^*^-^: partial rescue of scotopic ERG function	-wt mice: no retinal changes -*Rho^P23H^, Cep290^-/-^, Nr2e3^-/-^*: improved fundus exam findings -*Rho^-/-^, Rho^P23H^, Cep290^-/-^, Nr2e3^-/-^:* preserved retinal integrity	-*Rho^-/-^, Rho^P23H^, Cep290^-/-^, Nr2e3^-/-^:* increased rhodopsin and blue/green opsin expression; extensive gene expression changes

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
