# Peer review of "Targeting of the NRL Pathway as a Therapeutic Strategy to Treat Retinitis Pigmentosa"

_jcm, 2020, doi:10.3390/jcm9072224_

Round 1
Reviewer 1 Report
This review article examines in detail apromising treatment strategy for retinitis pigmentosa (RP). RP is a diverse group of orphan retinopathies which constitutes one of the leading causes of blindness in developed countries. More than a hundred genes are implicated in RP. Therefore, there is great interest in the field in identifying common therapeutic strategies rather than developing treatments for each rare gene mutation. The authors present modulation of NRL and NR2E3, two key rod photoreceptor cell fate determination transcription factors, as a potential promising mutation-independent treatment paradigm. Roles of these factors in photoreceptor development, determined by studies on animal models, human mutation clinical presentation, and pre-clinical work in modulating NRL and NR2E3 are described. Since many gene therapies specific to individual mutated genes are in advanced development, discussion of potential mutation-independent approaches that would allow treating a greater number of thus far orphan conditions is of interest to the broader readership in vision research. The topic is very interesting, and the article is well written. However, the authors need to address the following to enhance the quality of the review:
Overall organization as well as flow of discussion in specific subsections require some improvement. The introduction describes in detail RP and potential general treatments but does not even introduce NRL and NR2E3. It might be beneficial to broadly define background to the specific topic and briefly describe the proposed approach. Discussion of the human presentation of NRL and NR2E3 mutations at the very end of main text seems counter-intuitive, this section would more logically precede discussion of animal models and potential therapeutic approaches, so that comments on how these relate to clinical features in patients can be made. Authors may also consider improving organization in subsections, for instance by first describing gene expression level changes and then phenotypes in functional readouts for each gene.
The authors have not included some of the more recent references on NRL and NR2E3. For example, ref 41 is at an odd place for NRL’s effect on gene regulation. The authors should also look at the following: PMIDs 15163632, 15459973, 15591106, 27880916, 21813673, 28863214.
The authors should discuss the differences between the models and the patients as far as NR2E3 and NRL pathways go. The fact that in humans the disruption of NR2E3 leads to enhanced S cone syndrome and unusual degeneration (PMID 15229190), which is similar to Nrl-knockout phenotype in mice where all rods become S-cones. In mice, Nr2e3 loss results in initial preservation of rods but they express cone genes eventually leading to degeneration (PMID 21813656).
Specific comments:
Line 18 ‘mutation-agnostic’ – mutation-independent might be better phrasing, especially for international audience
Line 20 ‘opsins’ – ‘opsin’ in singular would indicate a single blue light-sensitive opsin that is upregulated in these cells
Lines 59-62 – it should be mentioned that RPE65 gene is expressed in retinal pigment epithelium rather than photoreceptors
Line 70 ‘works’ – ‘work’ in plural
Line 71 – this is a too broad generalization, some autosomal dominant diseases may benefit from increased expression of the correct allele
Line 72 – style, replace ‘work’ with a different word
Line 75 ‘Conceptual’ – ‘Conceptually,’
Line 76-78 – no need to start with capital letters when listing within one sentence
Line 84 ‘CNTF-eluting’ – ‘CNTF-releasing’
Lines 100-111 – brief description of preclinical transplantation of photoreceptor cells would improve this section, especially into RP animal models
Lines 124-125 – ‘the visual resolution outcomes of such an approach are unproven’ – specify that visual resolution obtained may be limited
Line 132 ‘NRL and NR2E3 nuclear receptors’ – NRL is not a nuclear receptor, should be ‘transcription factors’
Line 133 ‘discovered by Anand Swaroop’s group’ – Dr Swaroop cloned and characterized NRL. NR2E3 was identified independently by other groups.
Line 137 ‘maintaining rod photoreceptor differentiation’ – please consider rephrasing, for instance to ‘implicated in differentiation and subsequent maintenance throughout lifespan.’
Line 138 - either both gene names should be in italic or both in normal script
Line 140 – this phenotype was confirmed on physiological but not molecular level, please state
Line 141 ‘NRL/NR2E3 pathway’ – authors actually mean ‘blocking NRL/NR2E3 pathway’
Lines 141-142 ‘which may make these cells resistant to the cell death which occurs in RP.’ – many RP genes are specifically expressed in mature rods, downregulation of a rod-specific transcriptional program by blocking NRL presumably also prevents manifestation of disease phenotypes and this way prevents cell death, it’s a different mechanism than resistance to cell death
Line 145 - the schematic could be reorganized to show that Otx2 is the first photoreceptor transcription factor expressed, then Crx and Nrl are induced and often cooperate in triggering expression of rod genes, whereas Nr2e3 is primarily involved in suppression of cone genes
Line 157 - it should be clarified that Nrl knockout animals were instrumental in identifying S cone as the 'default' photoreceptor fate. Trb2 is another relevant gene. See also PMID 21813673
Line 159 - rod-cone intermediates are characteristic of Nr2e3 loss of function retina not Nrl knockout retina. In Nrl knockout, photoreceptor cells resemble S cones as far as it can be determined (see PMID 27326930)
Line 160 – ‘increased’ should be removed in this case it is either a cone or rod nuclear morphology not a gradual change
Line 161 – it should be stated that this was a response to blue light, red-green cone activity remained unchanged
Line 163 – Almost all RP genes are transcriptional targets of Nrl, not its interacting partners (though a few like CRX and NR2E3 are)
Line 165 – primary function described for Nr2e3 is suppression of cone gene expression and this should be highlighted in the title instead
Lines 176-180 – most definitive studies indicate that Nr2e3 is expressed downstream of Nrl in post mitotic rod precursors where it suppresses cone gene expression, not in proliferating retinal progenitors
Lines 182-185 – do the authors refer to autosomal dominant mutations in NRL and NR2E3? otherwise it does not logically follow that suppression of Nrl/Nr2e3 might provide any therapeutic benefit; better explanation in this section would be recommended
Line 208 – reduced or eliminated Nrl protein expression from photoreceptors, rather than knocked out (which can indicate germline)
Line 230 – most mutations in NRL causing RP are autosomal dominant, autosomal recessive mutations can lead to enhanced S cone syndrome, this needs to be clarified
Line 235 - it would be worth to hypothesize why deletion in adult mice has a different effect, one conceivable explanation is that epigenetic modifications on photoreceptor gene loci are established in development and remain fixed not allowing for conversion into cones like during development
Line 276 - there are two “in”.
Lines 311-313 - to many readers this will suggest that the strategy that authors propose might not be applicable to humans, authors should elaborate
Lines 321-322 – this is an imprecise statement, these therapies aim to bypass photoreceptors and enable vision using interneurons or ganglion cells which often persist in RP, real shortcomings of optogenetic strategies are poor sensitivity and resolution provided by the optogenetic tools
Lines 334-342 – downregulation of rod specific gene expression is the more likely explanation and would be better presented first
Lines 351-353 – clarification that the 2% authors mention is comparing developmental to adult Nrl knockout should be explicitly spelled out
Line 362 – replace one of the underlined ‘further’
Reviewer 2 Report
The authors gave a comprehensive description on targeting of the NRL pathway and its potential therapeutic use to treat retinitis pigmentosa. The manuscript is easily readable. Minor comments:
- On page 2, the authors wrote "viral vectors are limited in carrying capacity (maximum of 7.5kb for recombinant AAV) of cDNA" which is wrong. Should be corrected to a maximum of 4.8 kb (including the two inverted terminal repeats) for recombinant AAV.
- The authors well described the potential use of targeting the NRL pathway to turn non-viable rods into viable cone photoreceptors. But the text would gain by a few sentences on potential positive as well as any potential negative outcome for the patients. The human retina preferentially does contain rods, in addition to several subtypes of cones, for optimal function. In addition, germline patient mutations in the NRL pathway cause retinal degeneration, whereas Nrl depletion in adult mice do not cause retinal degeneration. Morphological, retinal function as well as behavioral tests would be required before clinical application of inhibition of the NRL pathway in NRL pathway mutant retinas.
Reviewer 3 Report
The manuscript is well written and well presented, although it could be enriched with some information relating to the importance of identifying and using specific biomarkers.
Section "1.2. Genetic Heterogeneity of Retinitis Pigmentosa" need to be improved. It is recommended to insert the main candidate genes/genetic variants involved in the etiopathogenesis of RP. In fact, the identification of specific genetic variants can also explain the onset of some phenotypes, such as the late onset form. In addition, it may be helpful to insert a section on genomic biomarkers, focusing on their usefulness in the clinical practice. In the literature there are numerous scientific papers, conducted on different pathologies (RP, AMD, Psoriasis, Parkinson disease, cancer), which focus on this topic (Ly A, et al., Clin Exp Optom; Stocchi L et al., Curr Genomics; Docampo E et al., Arthritis Rheum; Cascella R et al., Pharmacogenomics; Couñago F et al., Cancers (Basel)). Moreover, the utility of having specific biomarkers available, can also be included in the conclusions.
Round 2
Reviewer 3 Report
The manuscript is correctly revised.